# The Energy Impact of Building Materials in Residential Buildings in Turkey

**DOI:** 10.3390/ma14112793

**Published:** 2021-05-24

**Authors:** Pınar Usta, Başak Zengin

**Affiliations:** 1Department of Civil Engineering, Technology Faculty, Isparta University of Applied Science, Isparta 32200, Turkey; 2Department of Civil Engineering, Faculty of Architecture and Civil Engineering Nisantasi University, Istanbul 34000, Turkey; zenginbasak@gmail.com

**Keywords:** residential building performance, energy efficiency, cost efficiency, building material

## Abstract

In Turkey, heat loss from existing and new buildings constitutes a large part of energy waste, so usage of suitable construction material is quite important. The building selected in this study was analyzed by applying different building materials considering the annual energy consumption allowed, and according to the different heat zones and different thicknesses of insulation material in relation to demand. The most suitable building material in terms of energy and cost uptake and cost given to the regions was determined; the results were measured in the study in terms of the maximum allowable annual heating energy requirement and the optimum values were determined. Comparison of the optimum values and the total energy consumption rates was conducted for the analyzed cities.

## 1. Introduction

Climate change is a tremendous long-term challenge facing the Earth today. The energy- and thermal-performance of buildings has gained global importance in recent years, due to the aim of maintaining thermal comfort with a more efficient approach [1,2].

It is reported that public and commercial buildings in Europe consume an estimated 40% of total energy [3]. Residential buildings alone represent most parts of the final energy demand in many rural areas, which makes them one of the major single energy consumers of residential buildings. However, the operational energy demand, with a specific emphasis on thermal aspects, seems to cover a large part of the overall energy consumption of residential buildings and their users [4]. The construction sector must take responsibilities for environmental problems, as in every phase of the construction life-circle energy is consumed at a different level. Construction materials represent an important share of this consumption, and the energy consumed by the building materials during their life cycle becomes a significant parameter in the determination of the energy efficiency of the construction [5].

All building materials in their life cycle are exposed to different negative factors that influence their durability. One of them is carbon dioxide (CO_2_) [6]. The building construction industry uses a lot of energy and emits large amounts of carbon dioxide (CO_2_) into the atmosphere. Energy is used to extract, transport, process and combine materials, and CO_2_ is released into the environment through fossil fuel burning, land use applications, and industrial process reactions. Recently, increased knowledge has suggested that building with Aerated Autoclaved Concrete (AAC) and wood-based material can result in lower CO_2_ emissions compared to other materials, such as concrete, brick, or steel [7]. Concrete is the most widely used construction material in the world, with the prevailing consumption of 1 m^3^ per person per year [8]. A variety of factors affect the energy and CO_2_ balances of building materials over their lifecycle. The sustainable development of concrete construction requires sustainable materials, sustainable binder, low carbon trace, and minimum CO_2_ emissions, etc. These requirements can only be fulfilled if the rules for the optimal design of the materials are observed [9].

Brick is one of the most important building materials used in construction to make walls and pavements. However, the continued use of clay bricks in the construction industry leads to widespread loss of fertile topsoil, which can pose an environmentally devastating hazard. Near a brick kiln, environmental pollution from brick making harms health, animals, plant life and causes a number of environmental and health problems. Environmental pollution caused by the brick production process contributes to global warming and climate change. Moreover, air temperature can cause brick surface degradation due to frost damage, leading to global warming concerns. Various types of blocks are used as an alternative to red bricks to reduce environmental pollution and global warming problems. Aerated concrete blocks can be one of the solutions to replace traditional clay bricks and pumice. The manufacturing process of autoclaved aerated concrete (AAC) blocks does not cause any environmental problems. Furthermore, AAC is a certified green building material, which is porous, nontoxic, reusable, renewable and recyclable which can be used for commercial, industrial and residential construction [10].

There are two main concerns in the energy category which need to be addressed. One of these concerns is the improvement in CO_2_ emissions compared to the current Building Regulations standard. The purpose of this credit category is to minimize CO_2_ emissions into the atmosphere from the operation of a house and its services, and this is done by assessing the amount of CO_2_ emitted from the dwelling as a result of space heating, hot water, and lighting. The second section of credits is aimed at future-proofing the energy efficiency of dwellings over their whole life by limiting heat losses across the building envelope. Improving the basic thermal performance of the fabric of a house is a major step towards meeting the higher Code levels. The thermal performance of AAC can contribute considerably in this respect and help reach higher Code credits. Cost-effective AAC external walls can be constructed to achieve a wide range of U values in both cavity and solid wall construction [11].

There are relatively few studies that analyze the effect of carbonation on matrices in composites based on mineral binders, including AAC. However, the studies with CO_2_ concentration for 14 days have shown that all samples of AAC are resistant to carbonation. Tested samples from different producers and from different countries have shown that AAC is a durable material, independent of the process technology used [6].

Energy-efficient materials can sustain construction both ecologically and economically because of their environmentally friendly features. Furthermore, energy-efficient materials, with their various thermal properties, contribute to comfortable indoor environments [12]. 

In many countries, the energy requirement for space heating in buildings has the highest parcel of all, which is about 40% of total energy consumed in the residential sector. Regarding energy consumption, heating accounts for the largest share in the residential and tertiary sector in Greece (60.9% and 52.5%, respectively). This is an average of 57% in the European Union. It is clear from the data above that effective thermal protection in the residential sector plays a significant role in the reduction of energy expenditure for space heating. Proper building material and efficient insulation thickness in buildings could be the most effectual way of energy conservation in residential applications. Choosing proper building material reduces fuel consumption, undesirable emissions from the combustion of fossil fuels, and increases thermal comfort by minimizing heat losses from buildings [13,14,15].

Residential energy use depends mainly on the available amounts of local resources, which are closely connected with the present rural economy and living standards [16]. Energy consumption for heating is also high in Turkey since many buildings have almost no insulation or inefficient insulation [17]. While designing and constructing the buildings, paying attention to the climatic characteristics of the region contributes to the total cost economy and efficiency of the building. Energy conservation to reduce lifecycle energy costs has also become an important consideration while designing buildings [18]. Building materials for energy efficiency have been contemplated by many researchers, of which some are related to this survey [19,20,21,22,23].

## 2. Material and Method

In this paper three different building materials are applied to the same 3 story residential buildings, which are in various climate zones of Turkey, to determine and compare energy validity. The methodological framework of the study is presented in Figure 1.

The energy performance of the dissimilar types of buildings, the calculation method of annual heating energy demand, thermal transmittance “U” values for each region, which is determined by using the “degree- day method” in TS-825, and the maximum heating demand values according to regions were reported. The maximum U-value requirements according to TS825 [24] are given in Table 1. After describing the maximum heating demand values, the monthly outdoor temperature and solar radiation, which were assumed into consideration in this measure to calculate heating loads of buildings, were classified separately according to each region and month. In addition, the maximum heating loads were applied according to the A/V (area/volume) rates of buildings for each region (Table 1) [21].

In this study, Antalya, Istanbul, Ankara and Erzurum—as four sample cities located in different thermal regions—are selected to demonstrate the effect of decisions regarding the thickness of insulation materials with various building materials. Istanbul and Ankara are the moderate cities in the second and third degree-day region, respectively. Turkey is carved up into four climatic regions depending on temperature degree-days of heating according to the Turkish thermal insulation regulations (TS 825) given in Figure 2 [15,25,26].

The selected cities and their data are depicted in Table 2. Additionally, the energy savings and cost analysis resulting from the use of building materials and insulation were compared at a base “Maximum Allowable Annual Energy Requirement”, which were estimated individually for each region.

When the annual average temperature data in Antalya is examined, it is found that out that the annual maximum average temperature is 24.1 °C. The average maximum temperature values of the months range between 14.9 and 34 °C. The average maximum temperature difference between Antalya’s hottest month and coldest month is 20.1 °C (Figure 3).

When the yearly average temperature data in Istanbul is examined, it is seen that the annual maximum average temperature is 18.7 °C. The average maximum temperature values of the months range between 8.8 and 28.9 °C. The average temperature difference between Istanbul’s hottest month and the coldest month is 19.1 °C (Figure 4).

When the annual average temperature data in Ankara is examined, the annual maximum average temperature is seen as 17.8 °C. The normal maximum temperature values of the months range between 4.1 and 30.4 °C. For that reason, the average temperature is in February (−2.3 °C) when the temperature is lower, and the average temperature of August is 30.4 °C. The average temperature difference between Ankara’s hottest month and coldest month is 26.3 °C (Figure 5).

When the year-round average temperature data in Erzurum is examined, it is shown to be that the annual maximum average temperature is 11.9 °C. The average maximum temperature values of the months range between −4 °C and 27.2 °C. The maximum average temperature difference between Erzurum’s hottest month and coldest month is 28.5 °C (Figure 6).

## 3. Description of the Building and Building Material

The studied building is a 3-story residential building, which consists of the ground floor and three floors which has a gross area of about 910 m^2^. Each unit has two bedrooms, one living room, one kitchen and one bathroom. The building has two dwelling units on each floor with about 455 m^2^ and the floor plan of the house is shown in Figure 7. The structure of the building envelope components is shown in Table 3.

The systems (Table 3) and costs are similar in all regions of Turkey for products easily available in a traditional structure. The systems discussed in the study are oriented toward providing the best, most suitable and most economical exterior wall structure system according to the best annual heating energy demand value. The m^2^ design and all features of the building are the same and the annual heating energy demand values are equal.

There is no demand for insulation AAC systems in the 1st- and 2nd-degree regions, the wall provided the needed values in the regulation (TS 825), nevertheless, insulation was required in the 1st, 2nd-zone brick, and pumice systems and all the systems in the 3rd and 4th zones.

### 3.1. Calculation of Annual Heating Energy Data of Building

In the study, the optimum annual heating requirement rate was determined for all external wall systems. Temperature loss and energy demand of cities belonging to four different degree day zones are calculated monthly according to TS 825 regulation. In the calculations, the quantity of fuel to be consumed per unit volume or unit area (kg.m^3^) 860 × Q_year_/(calorific value of fuel × system efficiency) (Kcal/kg.m^3^) = 1.17 (kg.m^3^) was taken as fuel. The annual heating energy demand for a single building section in buildings is calculated with the following equation:Q_year_ = ∑Q_months_(1)
Q_months_ = [H (θ_i_ − θ_e_) − η_ay_ (ϕ_i,ay_ + ϕ_s,ay_)]·t (2)
where; Q_year_: annual heating energy (Joule), Q_months_: monthly heating energy (Joule), H: specific heat loss of the building (W/K), θ_i_: average monthly internal temperature (°C), θ_e_: average monthly outside temperature (°C), ηmonths: monthly average usage factor for earnings (unitless), ϕ_i, months_: monthly average earnings (can be received fixed) (W), ϕ_s, months_: monthly average solar energy gain (W), t: time, (a month in seconds = 86,400 × 30) (s)All calculation results represented by graphics, seen in (Figure 8, Figure 9, Figure 10, Figure 11, Figure 12 and Figure 13).

As it is understandable from the figures, the heat demand in the four different degree day regions decreases towards summertime months, thus the heat loss decreases in proportion to the summer months on aforesaid dates. At the beginning of the subject, since the annual heat requirement values of the building materials are optimized by calculations and this value is kept constant and equal, the heat demand and heat losses of the materials are approximately the same in the graphics. Climatically, the 1st degree day zone is the lowest in terms of heating need and the highest in terms of cooling need. For this reason, Erzurum province, selected in the study, gives the highest heat demand and heat loss among other provinces (Antalya, İstanbul and Ankara). In areas with a high cooling load, proper insulation will provide long-term economy and comfort in the structure in terms of the operating period.

### 3.2. Condensation and Evaporation Amounts in Building Elements

In buildings that are situated in countries where external ambient temperatures vary in a wide range, such as Turkey, the importance of material insulation applications are increasing day by day in order to bring down the heat losses in winter months and the heat gains in summer months. The condensation that occurs as an outcome of water vapor diffusion negatively affects the heat transfer occurring in building materials Condensation or perspiration on surfaces that occur in building materials, especially in cold seasons, change the physical and thermal properties of building materials. As a result, condensation increases the overall heat transfer coefficient of the material, to the point that it can disrupt the structure of the material and cause increased heat loss. This phenomenon, which is called condensation or sweating, causes undesirable outcomes such as damage to the materials, reduced intensity levels and increased heat losses due to the increased overall heat transfer coefficient. Condensation occurs due to a lack of insulation or insufficient insulation [28,29,30,31,32,33]. The condensation and evaporation amount of building materials according to the climatic zones are presented in Figure 14, Figure 15, Figure 16, Figure 17, Figure 18, Figure 19, Figure 20, Figure 21, Figure 22, Figure 23, Figure 24 and Figure 25.

Commensurate to all results obtained from the analysis, no condensation occurred in the building element according to TS 825. Since the temperature deviation between the inner surface and the indoor environment is less than 3 degrees in exterior walls made using AAC, it shows conformity with the ordinances.

According to the results obtained from the analysis, no condensation occurred in the building element according to TS 825. Since the temperature difference between the inner surface and the indoor environment is less than 3 degrees in exterior walls made using AAC, it follows the regulations.

Appropriate to the solutions obtained into the analysis, no condensation occurred in the building element according to TS 825. Since the temperature difference between the inner surface and the indoor environment is less than 3 degrees in exterior walls made using AAC, which follows with the regulations.

Condensation conditions have occurred in one component in the building element. Since the temperature difference between the inner surface and the indoor environment is less than 3 degrees, this is in compliance with the Standard. The quantity of condensed water in the 3rd component is higher than the limit specified in TS 825 of 5.44 > 0.5 kg/m^2^. Condensation occurred in the heat, waterproofing or air layer (Max. 0.5 kg/m^2^). The mass of the condensed water is less than the mass of the evaporated water; hence, the condensation is harmless.

Characterizing the results obtained from the analysis, no condensation occurred in the building element according to TS 825. The temperature difference between the inner surface and the indoor environment is less than 3 degrees in exterior walls made using AAC, which is in observance with the requirements.

### 3.3. Cost Analysis

According to the results obtained from the analysis, no condensation occurred in the building element according to TS 825. Since the temperature difference between the inner surface and the indoor environment is less than 3 degrees in exterior walls made using AAC which compliance with the regulation. Energy saving to wall unit m^2^ costs (USD) given in Table 4.

The construction of the configuration selected to study and the three different materials in Turkey were evaluated according to four distinct values that meet the demanded TS 825 regulations. Attuned to the results, despite that 10 cm thick insulation material was applied on the wall materials in Zone 4, brick and pumice products could not conform to the limit values defined in TS 825. In the 1st- and 2nd-degree day zones, AAC provide the desired values without the need for any extra insulation material, and the demand limit values in the 3rd and 4th-degree sections were achieved by putting on an additional insulation material layer.

## 4. Results and Discussion

While designing a project/building, the standards, manufacturing terms, price, mechanical and physical properties of the chosen material are important. Eco-friendly production has also become essential due to recently increasing levels of global warming and the environmental problems which come with this.

The energy parameters of the preferred materials in building design are taken into consideration. In a redesigned structure in Turkey or anywhere in the world, the basic principles of building design are similar, except for the environmental conditions. When calculating the cost, the choice of materials that meet the standards is reduced. Brick, pumice, and cement-based blocks have high levels of CO_2_ emissions during production, which causes global warming levels to increase. In addition, the quality and standard for these materials vary from country to country, because the ambient conditions affect the setting process of the concrete during the transition of the material from the plastic state to the hardened state and the quality of the concrete changes. In the production of concrete and pumice, ambient conditions and even the source differences of the raw materials in the mixture change the quality of the material. AAC production requires a serious investment and a technical infrastructure all over the world, so almost the same standard and quality can be provided globally.

Different insulation thicknesses have been added to external walls in order to progress to TS 825 regulations in zones, where the annual heat values of building materials do not demand a limit value. Even though 10 cm thickness was added in the insulation material in some zones, the limit values are still not attained. The heat values calculated according to TS 825 annual heat amounts and materials are represented in Table 5.

According to the energy efficiency index of the building: if Q_year_/Q′ 0.99 or ≥0.90, that was classified as a C type building, if Q_year_/Q′ 0.90 or ≥0.80, a B type building, if Q_year_/Q′ 0.80 it is classified as an A-type building. Attuned to the standards of the thermal insulation rules in buildings, the annual heating requirement was estimated as “Q_year_” in line with the architectural characteristics and proportions of buildings. The annual heating energy demand should be smaller than standard limit values in line with the building architectural features and dimensions. Table 6, Table 7 and Table 8 express the limit values according to the regions and whether those values are sufficient or not.

The energy efficiency index of the building considered in the study, according to the materials can be seen in Table 9.

## 5. Conclusions

Turkey’s energy demand is increasing rapidly proportional to its rising population. Considering this situation, the limited energy resources of the country, and its dependence on foreign resources, energy-saving becomes more and more significant every day. In Turkey, heat losses from buildings are one of the primary sources of energy waste. Consequently, significant energy savings can be achieved by designing and constructing buildings with suitable building materials and insulation materials. The type and thickness of the building and insulation materials play an important role in the heating and cooling of buildings in terms of energy consumption. Using proper building and insulation materials will cut down energy usage in buildings. For the building material of the external wall, the cost analysis in terms of the various building materials is calculated for the four different climatic regions in Turkey.

The three most commonly used materials have been selected in our study. These materials, which are applied according to the standards, were analyzed depending on the environment variables and insulation materials. The prevalence of the materials used in the analysis in Turkey and around world are similar. As a result of this study, it has been revealed that AAC is used more widely in buildings where energy consumption is important. Less energy is used in the production of AAC when compared to brick and pumice building materials. In addition, evaluating in terms of recycling, sustainability, and energy, AAC, which uses fewer natural resources, provides an advantage. These parameters and our study will be supportive for housing production in environments similar to the conditions in Turkey, and also for any part of the world.

In this paper, calculations were made for four cities in different climate zones, and the following conclusions have been drawn based on these calculations. The EPS insulation material was selected for 19 cm horizontal perforated brick, 19 cm pumice block and 20 cm aerated concrete block wall materials, which are widely used in Turkey for building exterior wall structures. For the structures with different exterior wall materials, heat calculations were made according to four different regions, and it was checked whether the values obtained could fulfill the demand limit values in the TS 825 regulations. The building considered in the study had a gross usable area of 460 m^2^, consisting of six independent sections with three floors, and an external wall area of 320 m^2^. The building is a class 4A building, according to the definition of the Republic of Turkey Ministry of Environment and Urbanization. The Republic of Turkey Ministry of Environment and Urbanization determined the unit cost per square meter of 4 A type buildings as 198.72 USD/m^2^ for 2020. Taking our considered data, the cost of this building was estimated at USD 90.416 in this analysis.

At the end of the study, for a structure of this size it can be seen that the AAC wall product provided 2.8% less cost in total, as there was no need for an additional layer of thermal insulation material on the 1st- and 2nd-degree zones. Comparing alternative systems in the 3rd-degree zone, the AAC and insulation system was 0.36% more expensive than other systems. Even taking into account the application of 10 cm thick EPS insulation material on the wall layers in the 4th-degree zone, the brick and pumice systems could not meet the TS 825 demand limit values.

Generally, the cost of insulation is generally higher in cold regions than in warmer regions, but the payback time for insulation is much shorter. When this is taken into consideration, short-term investments could reduce Turkey’s dependence on limited fuel sources and make important contributions to the Turkish economy, and thus considerable energy savings can be obtained by using proper materials in buildings.

## Figures and Tables

**Figure 1 materials-14-02793-f001:**
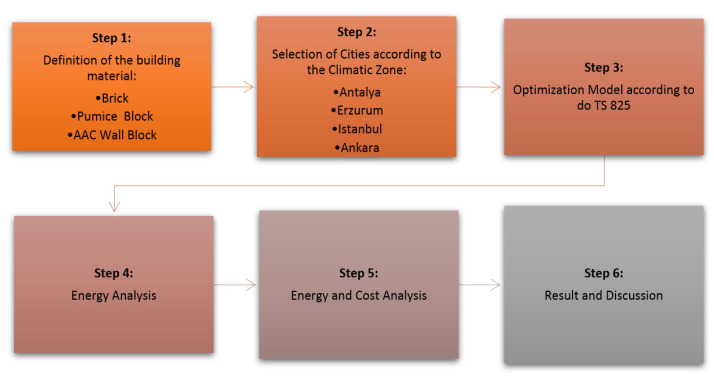
Methodological framework.

**Figure 2 materials-14-02793-f002:**
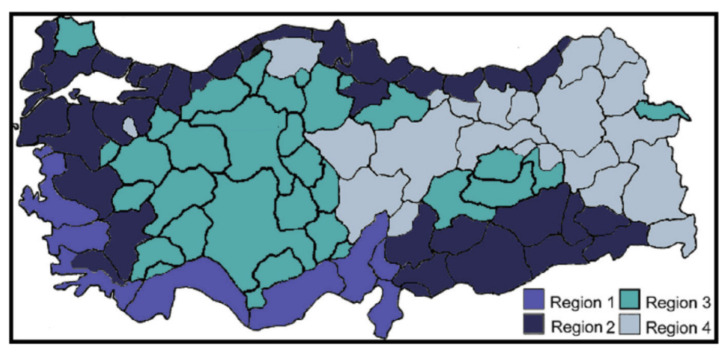
Four different degree-day regions of Turkey according to TS 825 regulations.

**Figure 3 materials-14-02793-f003:**
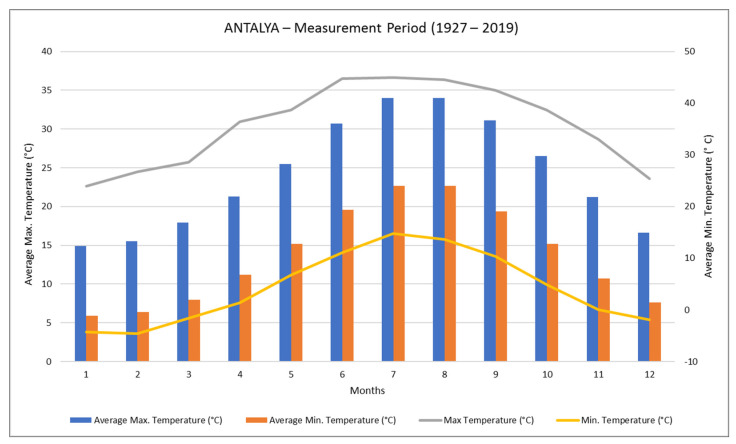
Annual temperature values of the Antalya (TSS 2021).

**Figure 4 materials-14-02793-f004:**
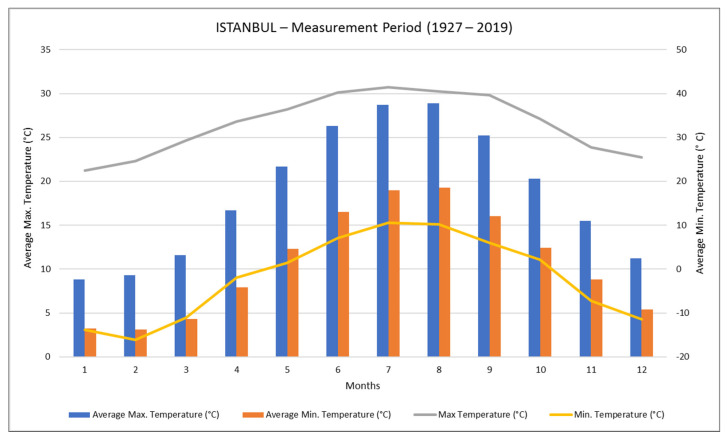
Annual temperature values of the Istanbul (TSS 2021).

**Figure 5 materials-14-02793-f005:**
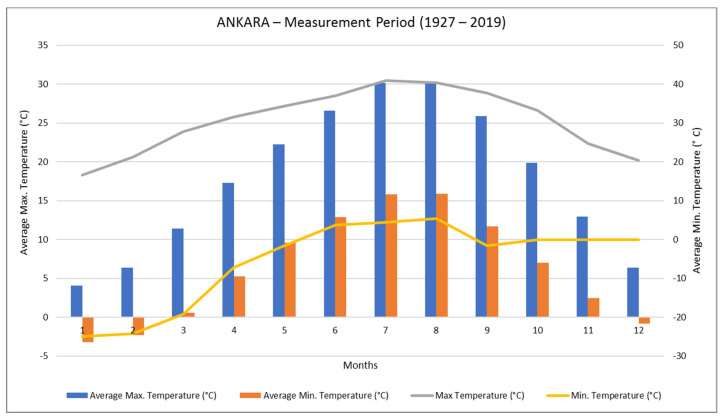
Annual temperature values of the Ankara (TSS 2021).

**Figure 6 materials-14-02793-f006:**
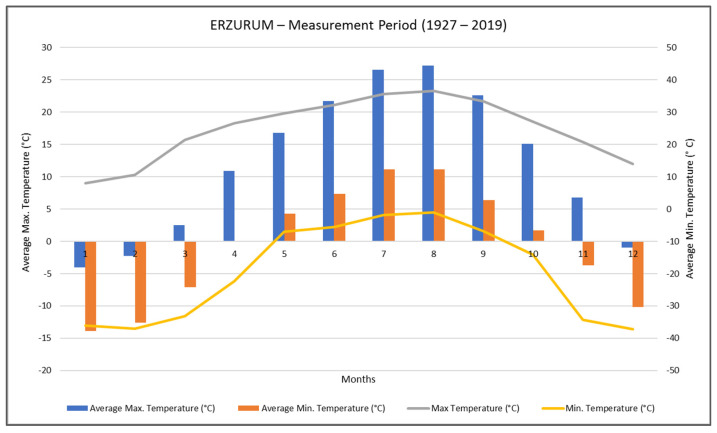
Annual temperature values of the Erzurum (TSS 2021).

**Figure 7 materials-14-02793-f007:**
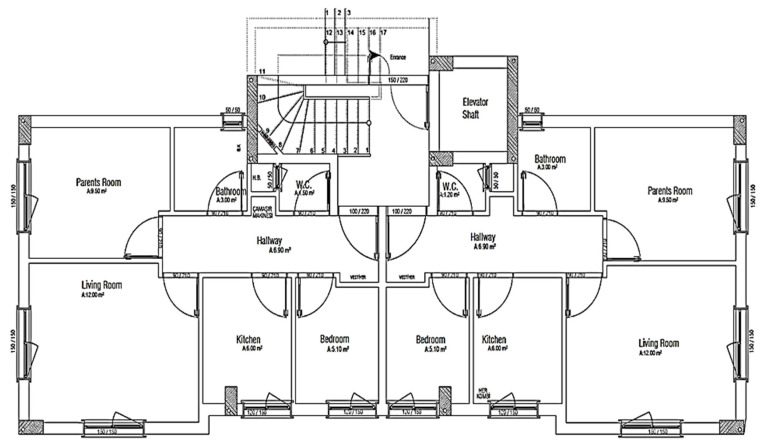
The first story plan of studied building.

**Figure 8 materials-14-02793-f008:**
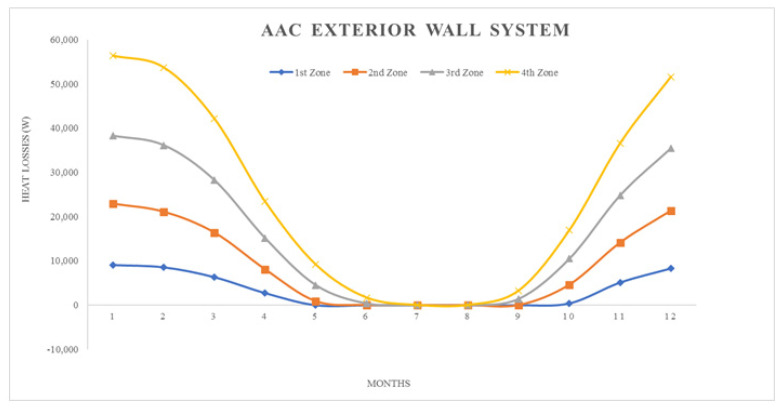
Heat losses for AAC exterior wall system.

**Figure 9 materials-14-02793-f009:**
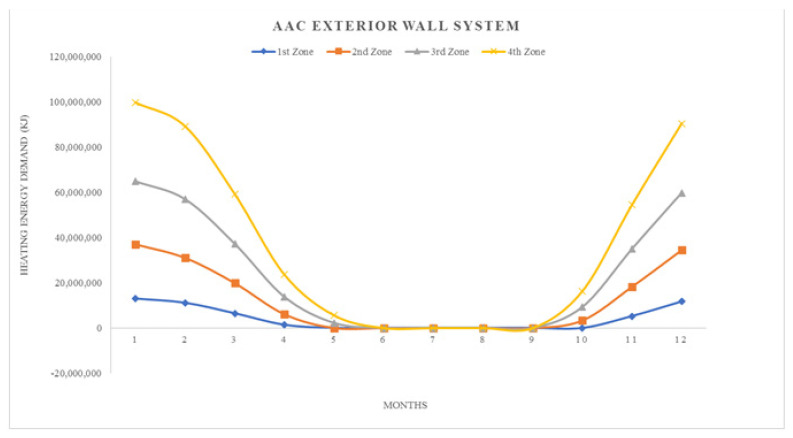
Heating energy demand for AAC exterior wall system.

**Figure 10 materials-14-02793-f010:**
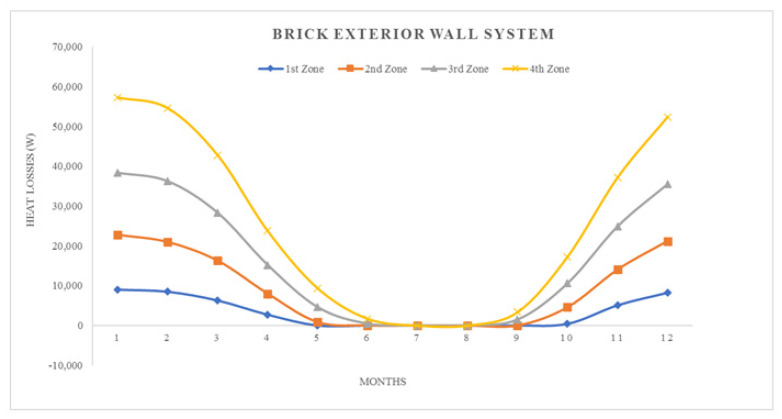
Heat losses for brick exterior wall system.

**Figure 11 materials-14-02793-f011:**
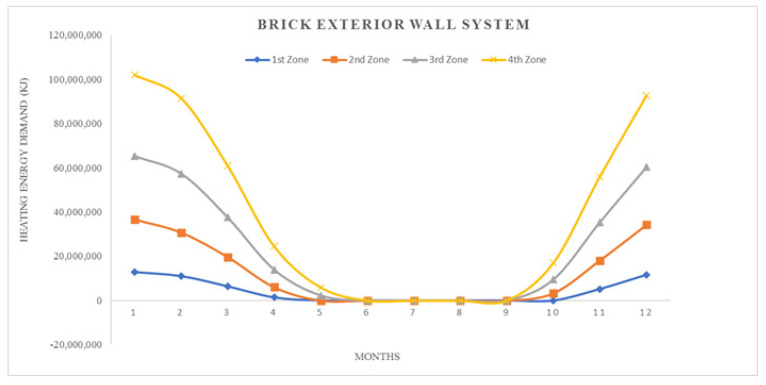
Heating energy demand for brick exterior wall system.

**Figure 12 materials-14-02793-f012:**
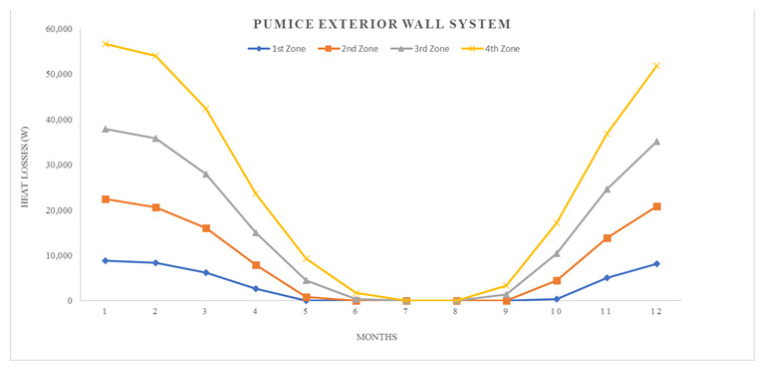
Heat losses for pumice exterior wall system.

**Figure 13 materials-14-02793-f013:**
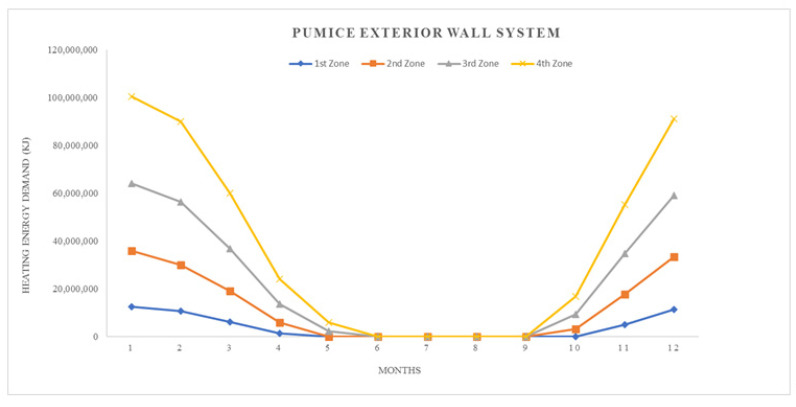
Heating energy demand for pumice exterior wall system.

**Figure 14 materials-14-02793-f014:**
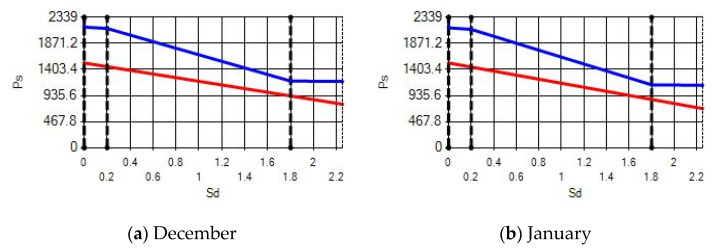
Condensation and evaporation amount chart for the building elements of the AAC external wall for the 1st zone for (**a**) December and (**b**) January.

**Figure 15 materials-14-02793-f015:**
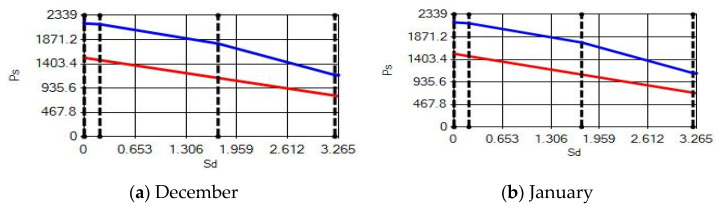
Condensation and evaporation amount chart for the building elements of the AAC external wall for the 2nd zone for (**a**) December and (**b**) January.

**Figure 16 materials-14-02793-f016:**
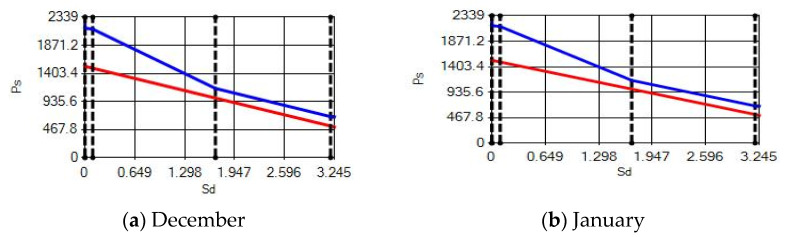
Condensation and evaporation amount chart in the building elements of the AAC external wall for the 3rd zone for (**a**) December and (**b**) January.

**Figure 17 materials-14-02793-f017:**
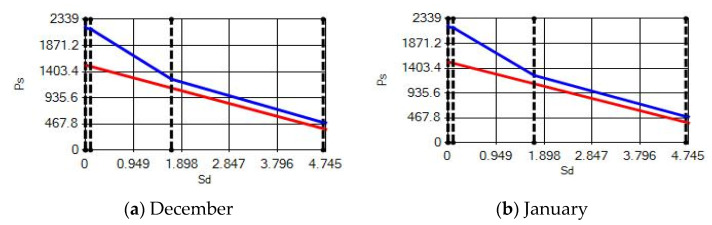
Condensation and evaporation amount chart for the building elements of the AAC external wall for the 4th zone for (**a**) December and (**b**) January.

**Figure 18 materials-14-02793-f018:**
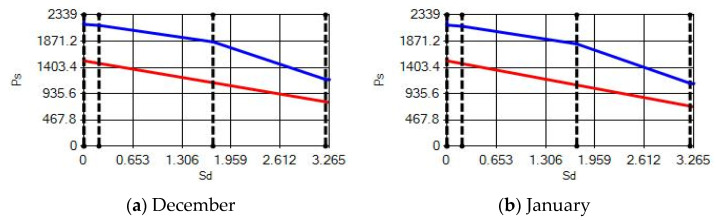
Condensation and evaporation amount chart in the building elements of the brick external wall for the 1st zone for (**a**) December and (**b**) January.

**Figure 19 materials-14-02793-f019:**
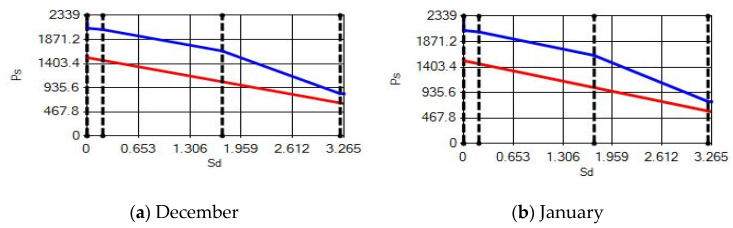
Condensation and evaporation amount chart for the building elements of the brick external wall for the 2nd zone for (**a**) December and (**b**) January.

**Figure 20 materials-14-02793-f020:**
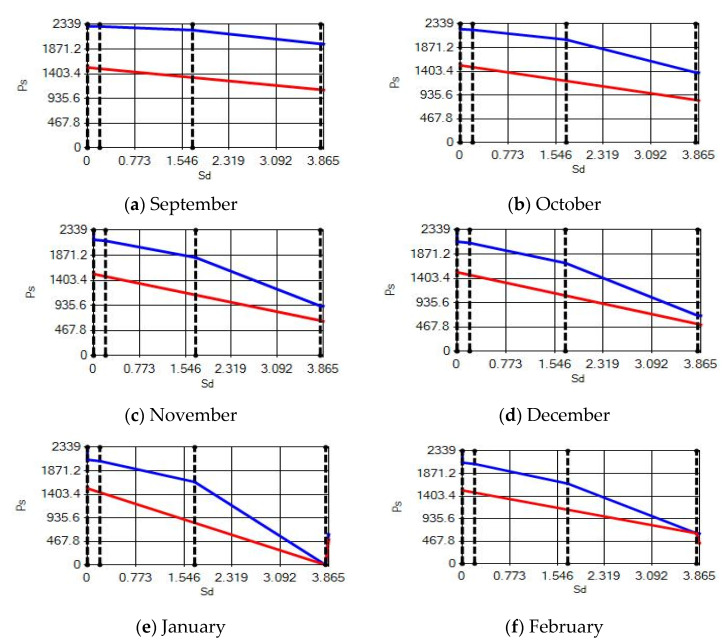
Condensation and evaporation amount chart in the building elements of the brick external wall for the 3rd zone for (**a**) September, (**b**) October, (**c**) November, (**d**) December, (**e**) January and (**f**) February.

**Figure 21 materials-14-02793-f021:**
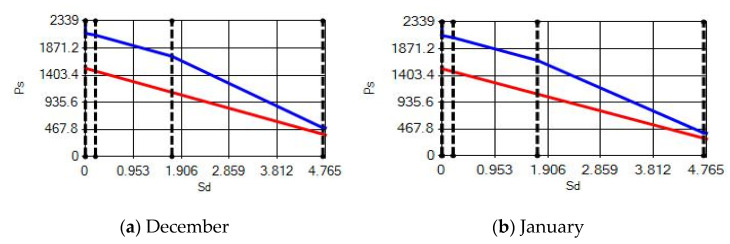
Condensation and evaporation amount chart in the building element of the brick external wall for the 4th zone for (**a**) December and (**b**) January.

**Figure 22 materials-14-02793-f022:**
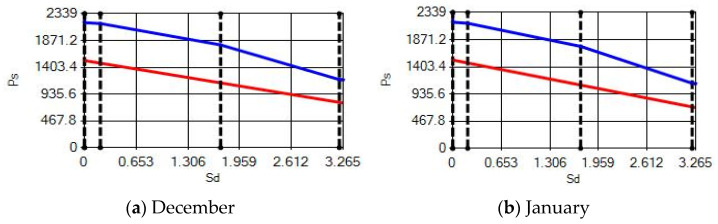
Condensation and evaporation amount chart for the building elements of the pumice external wall for the 1st zone for (**a**) December and (**b**) January.

**Figure 23 materials-14-02793-f023:**
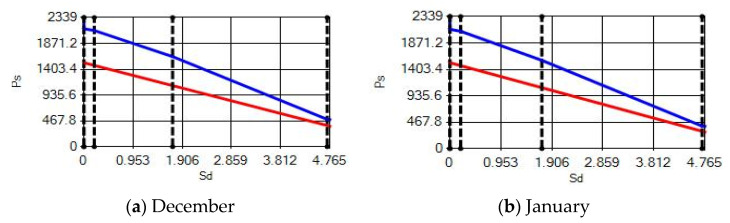
Condensation and evaporation amount chart for the building elements of the pumice external wall for the 2nd zone for (**a**) December and (**b**) January.

**Figure 24 materials-14-02793-f024:**
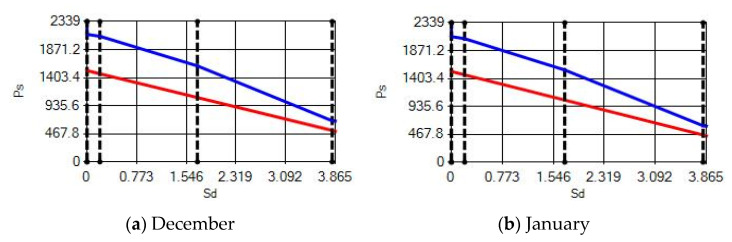
Condensation and evaporation amount chart for the building elements of the pumice external wall for the 3rd zone for (**a**) December and (**b**) January.

**Figure 25 materials-14-02793-f025:**
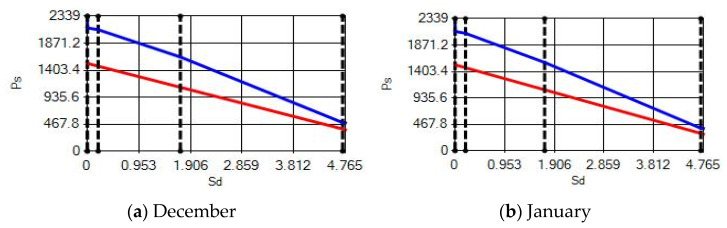
Condensation and evaporation amount chart in the building elements of the pumice external wall for the 4th zone for (**a**) December and (**b**) January.

**Table 1 materials-14-02793-t001:** Maximum U-value requirements (TS 825).

Climate Zone (TS 825)	Wall U (W/m^2^ K)	Roof U (W/m^2^ K)	Floor U (W/m^2^ K)	Window U (W/m^2^ K)
1st zone	0.7	0.45	0.70	2.4
2nd zone	0.6	0.40	0.60	2.4
3rd zone	0.5	0.30	0.45	2.4
4th zone	0.4	0.25	0.40	2.4

**Table 2 materials-14-02793-t002:** Data for selected cities [27,28].

City	Climate Zone (TS 825)	Altitude (m)	Longitude ^(^°^)^	Latitude ^(^°^)^	HDD	CDD	Degree-Days (°C Days)	Global Horizontal Radiation (kWh/m^2^y)
Antalya	1st zone	47	30,042′	36,053′	972	3345	1083	1798
İstanbul	2nd zone	33	29,005′	29,005′	1886	2152	1865	1465
Ankara	3rd zone	891	32,052′	39,056′	3307	1338	2677	1417
Erzurum	4th zone	1860	41,017′	39,055′	4785	856	4827	1555

**Table 3 materials-14-02793-t003:**
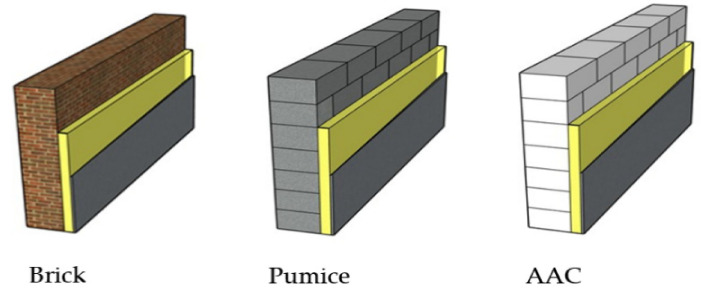
Structure of building envelope components.

	Material Thickness (cm)	Insulation Thickness	Material Thickness (cm)	Insulation Thickness	Material Thickness (cm)	Insulation Thickness
1st Zone	19	5 cm EPS Insulation	19	5 cm EPS Insulation	22	No need Insulation
2nd Zone	19	5 cm EPS Insulation	19	5 cm EPS Insulation	22	No need Insulation
3rd Zone	19	7 cm EPS Insulation	19	7 cm EPS Insulation	22	5 cm EPS Insulation
4th Zone	19	10 cm EPS Insulation	19	10 cm EPS Insulation	22	10 cm EPS Insulation

**Table 4 materials-14-02793-t004:** Energy saving to wall unit m^2^ costs (USD).

	Brick		Pumice		AAC	
	Material Thickness (cm)	Insulation Thickness	Material Thickness (cm)	Insulation Thickness	Material Thickness (cm)	Insulation Thickness
1st Zone	19	5 cm EPS Insulation	19	5 cm EPS Insulation	20	No need Insulation
Price	**16.02 USD**		**16.01 USD**		**8.59 USD**	
2nd Zone	19	5 cm EPS Insulation	19	5 cm EPS Insulation	20	No need Insulation
Price	**16.02 USD**		**16.01 USD**		**8.59 USD**	
3rd Zone	19	7 cm EPS Insulation	19	7 cm EPS Insulation	20	5 cm EPS Insulation
Price	**17.30 USD**		**17.43 USD**		**18.46 USD**	
4th Zone	19	10 cm EPS Insulation	19	10 cm EPS Insulation	20	10 cm EPS Insulation
Price	**19.87 USD**		**20.00 USD**		**21.02 USD**	

**Table 5 materials-14-02793-t005:** The 1st climate zone.

	External Wall System	Calculated Energy Requirement	Limits	TS 825 Condition
		Q (kWh/m^3^)	Q″ (kWh/m^3^)	Q < Q″
1	19 cm Brick + 5 cm EPS	9.88	12.34	Provided
2	19 cm Pumice + 5 cm EPS	9.65	12.34	Provided
3	20 cm AAC	9.98	12.34	Provided

**Table 6 materials-14-02793-t006:** The 2nd climate zone.

	External Wall System	Calculated Energy Requirement	Limits	TS 825 Condition
		Q (kWh/m^3^)	Q″ (kWh/m^3^)	Q < Q″
1	19 cm Brick + 5 cm EPS	20.42	21.99	Provided
2	19 cm Pumice + 5 cm EPS	19.96	21.99	Provided
3	20 cm AAC	20.62	21.99	Provided

**Table 7 materials-14-02793-t007:** The 3rd climate zone.

	External Wall System	Calculated Energy Requirement	Limits	TS 825 Condition
		Q (kWh/m^3^)	Q″ (kWh/m^3^)	Q < Q″
1	19 cm Brick + 7 cm EPS	27.00	27.16	Provided
2	19 cm Pumice + 7 cm EPS	26.62	27.16	Provided
3	20 cm AAC + 5 cm EPS	26.62	27.16	Provided

**Table 8 materials-14-02793-t008:** The 4th climate zone.

	External Wall System	Calculated Energy Requirement	Limits	TS 825 Condition
		Q (kWh/m^3^)	Q″ (kWh/m^3^)	Q < Q″
1	19 cm Brick + 10 cm EPS	34.32	33.09	Not Provided
2	19 cm Pumice + 10 cm EPS	34.04	33.09	Not Provided
3	20 cm AAC + 10 cm EPS	32.47	33.09	Provided

**Table 9 materials-14-02793-t009:** Building energy efficiency index.

		A Type	B Type	C Type
AAC Exterior Wall System	Antalya		x	
	İstanbul			x
	Ankara			x
	Erzurum			x
Brick Exterior Wall System	Antalya		x	
	İstanbul			x
	Ankara			x
	Erzurum			Need an extra layer to meet the TS 825 regulations
Pumice Exterior Wall System	Antalya	x		
	İstanbul			x
	Ankara			x
	Erzurum			Need an extra layer to meet the TS 825 regulations

## Data Availability

Not applicable.

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
