# Peer review of "The Energy Impact of Building Materials in Residential Buildings in Turkey"

_materials, 2021, doi:10.3390/ma14112793_

Round 1
Reviewer 1 Report
The manuscript presents an interesting case study in the field of energy efficiency. The manuscript addresses the issue of the energy impact of the building materials. The issue is being addressed in Turkey.
Figure 1 - Correct the picture (It is wrong to read the word Istanbul)
Figure 2 - The image is blurred. Improve quality. Do you have a copyright? Figure 3 and 4. Enlarge the image.
Figure 5 and 6 Enlarge the image.
Figure 7 Complete the diagram with the overall dimensions of the object. Variables are typically marked in italics - equations all, for example 2
Figure 8 and 9. Enlarge the image. There is no clear difference for
Figure 19 left and right. Is not it a mistake?
Figure 14, Figure 15, Figure 16, Figure 205 17, Figure 18, Figure 19, Figure 20, Figure 21, Figure 22, Figure 23, Figure 24, Figure 25. line 205 a 206 - Expand your comment and comment on the images in detail!
Figure 18. Why is the frame around the picture?
Table 9 - the table is on two pages. Reference 20 - add details to the citation. line 1 - correct the word Articl - Article Complete the section on copyright.
The manuscript must be processed with greater interest and focus on the visual manuscript. It would be appropriate to extend the addressed area by the area of evaluation (e-CO2 and e-Energy) or other areas. It is possible to use multi-criteria analysis for the evaluation Energy Impact. The manuscript must clearly present the new knowledge in the context of the journal Materials. The manuscript must be revised.
Author Response
For Command’s Reviewer 1,
The corrections requested by the first reviewer are arranged in the following order.
- First of all, the dimensions and images in the figures and tables were set up.
- The second, conclusion part has been improved and arrangements have been constructed according to modernization.
- Some arrangements were made by making references to new academic studies.
- As well, due to the version change of the doc file, a layout problem occurred.
- We are grateful for your attention and guidance, gained to improve our studies.
Figure 1 - Correct the picture (It is wrong to read the word Istanbul)
“I”, Istanbul text has been changed in Figure 1.
Figure 2 - The image is blurred. Improve quality. Do you have a copyright? Figure 3 and 4. Enlarge the image.
Figure 5 and 6 Enlarge the image.
Figure 8 and 9. Enlarge the image. There is no clear difference for
Figure 14, Figure 15, Figure 16, Figure 205 17, Figure 18, Figure 19, Figure 20, Figure 21, Figure 22, Figure 23, Figure 24, Figure 25. line 205 a 206 - Expand your comment and comment on the images in detail!
Changed in figure 2 and add a new figure which belongs to us so no need for copyright for this figure 2.
Changed dimensions and quality graphics, also get extra comment and details from images in Figure 5-25.
Figure 7 Complete the diagram with the overall dimensions of the object. Variables are typically marked in italics - equations all, for example, 2
Figure 19 left and right. Is not it a mistake?
Figure 18. Why is the frame around the picture?
Designed image newly and removed the frame.
Table 9 - the table is on two pages. Reference 20 - add details to the citation. line 1 - correct the word Article - Article Complete the section on copyright.
Checked references 20
Reviewer 2 Report
The article is interesting and correctly developed. What is missing is an indication of directions for further research at the end of the article.
Author Response
For Command’s Reviewer 2,
We are thankful for your involvement in our academic study. We have concluded the arrangements according to your and other evaluation reports.

Reviewer 3 Report
The paper aims at evaluating the energy consumptions applying different insulation materials in a specific case study. English must be revised by a mother tongue as there are several mistakes. The abstract is not focus on the main findings of the paper. The introduction is poor, it does not report the paper on the cost optimal approach for the insulation of buildings, showing the gap and the novelty of your paper. As it is, it seems not a innovative paper. Probably, the innovation is related to Turkey climate, but a comparison with other procedure is useful to improve the outcomes. Also the paper doi.org/10.3311/CAADENCE.1640 is useful as it reported similar considerations on different insulation materials for the building envelope in different climates. The methodology is very easy to understand? But it is not new. Show the difference between other methodologies and the requirements of the legislation in Turkey. Discussion and conclusion must be revised at all showing the most interesting aspects of your research.
Author Response
For Command’s Rewiever 3,
In the study, typographical mistakes and some term errors were compensated.
The publication suggested to us was examined and we involved to our publication according to that publication.
The conclusion part of the study has been developed.
First of all, thank you for your effort. Your suggestions for our study were really valuable to us.
Suggestion;
Also the paper doi.org/10.3311/CAADENCE.1640 is useful as it reported similar considerations on different insulation materials for the building envelope in different climates.
Read this paper and due to this proposal improved our paper.

Round 2
Reviewer 1 Report
The research area and results are from the context of the manuscript can better understand.
Thanks for the comments and manuscript edits.
However, the changes made are insufficient.
The research itself is interesting, but much information is already known. The solution is also very focused on Turkey.
It is necessary to present the translated research in the overall context of the solved problem.
It is appropriate to mention other options for solving The Energy Impact of The Building Materials, extending the assessment to the area of CO2-e or the use of alternative materials, etc. (in the introduction or discussion section)
Turner, L.K.et. al. Carbon dioxide equivalent (CO2-e) emissions: A comparison between geopolymer and OPC cement concrete. Constr. Build. Mater. 2013, 43, 125–130
Bilek, V. et.al. Frost Resistance of Alkali-Activated Concrete — An Important Pillar of Their Sustainability. Sustainability 2021, 13, 473.
Overall, it is necessary to increase the informative value and presentation of the manuscript.
I will ask the authors to focus on the introduction, discussion and conclusion.
The manuscript must be revised, after which it will be able to be published.
Author Response
Round ;
- Improved eco-friendly (CO2 emissions) introductions, results and conclusions sections.
- Assessment of materials CO2 of energy impact building materials that use in this paper.
- The energy materials solutions are also not only focused on Turkey, It is stated that paper is could use another country similar to Turkey geographic properties models
- To expand the article, the articles suggested by the referred and some different eco-friendly (CO2 emissions) papers are edit as references.
- All manuscripts revised and the changes showed another colour for round 1 and round 2
6.This article was extended with the paper of Turner et al and Bilek Turner et al (sustainability journal ) recommend by the referee.
- Editing the English language
- The file is deformed after loading due to different wherefore word versions
Upload last versions
Sincerely

Reviewer 3 Report
Sorry, but I can’t see any revision in the new version you provided. Pleas, add a new revision with the changes highlighted in another color. I see you comments but I can’t find them into the paper.
Author Response

(The authors gave the same response as above.)

Round 3
Reviewer 1 Report
The changes made the improvement of the manuscript.
The research area and results are from the context of the manuscript can better understand.
Overall, the authors improve the clarity and visual of the manuscript.
The results of the research and information value of the manuscript can be evaluated overall very well.
The manuscript can be published in the journal.
Author Response
Thank you for your comments
Reviewer 3 Report
Thank you very much for inserting in yellow the corrections. Several aspects have been considered, but the part on hygothermal performance was not adressed. I see you reply on this topic (Reply to the authors), but I can't find neither the comments and the refernce in the paper
Author Response
Since we studied the moisture part of Hygrothermal in our study, boundary and parameter data of TS825 were used and the analyzes were made according to the seasons and regions.
General information about Hygrothermal is given in the section "Condensation and Evaporation Amounts in Building Elements" using referances [28–33]. Afterwards, The Condensation and Evaporation Amount of Building materials according to the climatic zones are presented in Figure 14, Figure 15, Figure 16, Figure 17, Figure 18, Figure 19, Figure 20, Figure 21, Figure 22, Figure 23, Figure 24 and Figure 25. Since the analyzes and results are found as a result of our own calculations, there is no reference to any source in the study as this part is also in the discussion section.